# Microwave-Assisted Synthesis of 3-Hydroxy-2-oxindoles and Pilot Evaluation of Their Antiglaucomic Activity

**DOI:** 10.3390/ijms24065101

**Published:** 2023-03-07

**Authors:** Alexander M. Efremov, Olga V. Beznos, Roman O. Eremeev, Natalia B. Chesnokova, Elena R. Milaeva, Elena F. Shevtsova, Natalia A. Lozinskaya

**Affiliations:** 1Department of Chemistry, Lomonosov Moscow State University, Moscow 119991, Russia; 2Institute of Physiologically Active Compounds at Federal Research Center of Problems of Chemical Physics and Medicinal Chemistry, Russian Academy of Sciences (IPAC RAS), Chernogolovka 142432, Russia; 3Helmholtz National Medical Center of Eye Diseases, 14/19 Sadovaya-Chernogryazskaya St., Moscow 105062, Russia

**Keywords:** intraocular pressure, glaucoma, isatin, oxindole, microwave irradiation

## Abstract

Glaucoma is a widespread neurodegenerative disease for which increased intraocular pressure (IOP) is a primary modifiable risk factor. Recently, we have observed that compounds with oxindole scaffolds are involved in the regulation of intraocular pressure and therefore have potential antiglaucomic activity. In this article, we present an efficient method for obtaining novel 2-oxindole derivatives via microwave-assisted (MW) decarboxylative condensation of substituted isatins with malonic and cyanoacetic acids. Various 3-hydroxy-2-oxindoles were synthesized using MW activation for 5–10 min with high yields (up to 98%). The influence of novel compounds applied in instillations on IOP was studied in vivo on normotensive rabbits. The lead compound was found to reduce the IOP by 5.6 Torr (ΔIOP for the widely used antiglaucomatousic drug timolol 3.5 Torr and for melatonin 2.7 Torr).

## 1. Introduction

Glaucoma is the most frequent cause of irreversible blindness since it leads to the irreversible loss of retinal ganglion cells and optic nerve fibers [1,2,3]. Unfortunately, at the moment there are few effective pharmacological ways of treating glaucomatous optic neuropathy. The conventional, important, and commonly used way of therapy is a life-long daily use of hypotensive drugs to slow down the progression of optic nerve damage [4,5]. Surgical methods also do not provide a total cure. The therapeutic effect of surgery lasts from one to five years, depending on the type of glaucoma (open-angle or angle-closure) and the stage of the disease [6].

As a rule, the primary treatment of open-angle glaucoma is carried out through pharmacotherapy. The main risk factor for glaucoma is ocular hypertension, which is an increase in intraocular pressure (IOP). The increase in IOP leads to damage to the optic nerve head, ischemia, and the loss of ganglion cells in the retina. Therefore, most antiglaucomic drugs are aimed at lowering IOP. Currently, there are six classes of ophthalmic drugs used in the treatment of glaucoma: prostaglandin analogues [7], carbonic anhydrase inhibitors [8], beta-blockers [9], alpha-adrenergic agonists [10], miotics [11], and rho-kinase (ROCK) inhibitors [12]. The decrease in IOP occurs either due to a reduction in the production of aqueous humor by the ciliary body or due to an increase in uveoscleral outflow. Beta-blockers and carbonic anhydrase inhibitors work by the first mechanism, while prostaglandin analogs, miotics, and ROCK inhibitors work by the second. Alpha-adrenergic agonists reduce the secretion of aqueous humor and at the same time increase the outflow [13].

A limited number of IOP-influencing molecular targets leads to the fact that patients develop tolerance and drugs no longer work. For example, some beta-blockers have a 30% chance of tachyphylaxis or drug tolerance development in 1 month [14,15,16]. For other antiglautomatous drugs—up to 2–3 years for each new molecular target. Thus, the search for new compounds capable of lowering IOP to a safe level is an important and urgent task.

Recently, considerable attention has been paid to melatonin and dopamine receptors as promising targets in the search for antiglaucoma drugs [17,18], which causes interest in oxindole derivatives as well. Previously, we have observed that 2-oxindole derivatives can reduce IOP in animal models and have antioxidant properties [19,20,21,22,23]. The most promising were recognized as 3-hydroxy-2-oxindoles. The standard method for obtaining such compounds is the reaction of aldol condensation of the starting isatins with carbonyl and carboxyl compounds.

In general, the synthesis of several 3-cyanomethyl- and 3-carboxymethyl-3-hydroxyoxindoles is carried out by the condensation of isatins with cyanoacetic or malonic acids with simultaneous decarboxylation [24]. However, the long reaction time (3–72 h) prompted the search for a new efficient method for the synthesis of 3-hydroxy-2-oxindoles, containing cyano and carboxy groups in the side chain. For this purpose, the acceleration of the condensation of isatins with malonic acid derivatives under microwave irradiation was studied.

## 2. Results and Discussion

The starting isatins (**1b**,**c**,**e**,**h**) were obtained using the Sandmeyer reaction from the corresponding anilines. Nitroisatins (**1a**,**d**,**f**) were obtained by the nitration of isatin (**1g**) or 5-methoxyisatin (**1c**) according to the described procedures [19,22].

Condensation of isatins (**1a**–**h**) with cyanoacetic or malonic acids in the presence of an organic base under conventional reflux for 3 h leads to the formation of 3-hydroxy-3-cyanomethyloxindoles (**2a**–**h**) and 3-hydroxy-3-carboxymethyloxindoles (**3a**–**h**), respectively (Figure 1 and Table 1). We proposed that microwave irradiation can sufficiently increase the reaction rate.

The optimization of MW reaction conditions was performed using 5-nitroisatin **1a** and malonic acid (Table 2). The usage of other organic solvents, such as THF and EtOH, dramatically decreases the product yield. The piperidine and triethylamine were equally effective as base catalysts in this reaction, and the best results were obtained with 1.5 eq. of base in the case of malonic acid (see Table 2) and with 0.25 eq. of base for cyanoacetic acid.

The condensation of isatins (**1a**–**h**) with cyanoacetic and malonic acids proceeds with simultaneous decarboxylation by two alternative mechanisms, **A** and **B**. In the case of condensation with cyanoacetic acid, mechanism **A** is assumed (Figure 2), while condensation with malonic acid probably proceeds by mechanism **B** (Figure 3). Mechanism **A** involves first decarboxylation and the formation of acetonitrile enolate in situ, analogously to the condensation of aromatic beta-keto acids with isatins [25].

Mechanism **B** involves first the nucleophilic addition of malonic acid to isatin followed by decarboxylation [26]. This reaction mechanism indirectly confirms the fact that if the intensity of microwave irradiation is reduced, the product of the reaction of unsubstituted isatin **1g** with malonic acid will not be 2-(3-hydroxy-2-oxoindolin-3-yl)acetic acid (**3g**), but 2-(3 -hydroxy-2-oxoindolin-3-yl)propanedioic acid (**X**) (Figure 3).

Usually, for compounds **2** and **3**, the diastereotopically different CH_2_ protons are well resolved in 1H NMR spectra (see **2a**, Figure 1). Interestingly, in the case of 7-nitro or 4-nitrosubstituted cyanooxindoles, the diastereotopy in NMR was poorly resolved up to complete degradation into singlet for compounds **2b**,**d** (Figure 1).

The influence of all synthesized compounds **2** and **3** on IOP was tested in vivo using normotensive rabbits. According to the previous study of the dose-dependent influence of melatonin and 2-oxindole derivatives on IOP [19], the instillation of 50 µL of 0.1%wt. solution was chosen for the pilot study of the new compounds’ influence on IOP.

The obtained data are summarized in Table 3 and Figure 2 and Figure 3.

As can be seen in Table 3 and Figure 2, nitrile derivatives **2** are more active than acids **3**, which have the same substituents in the 2-oxindole ring. 5-Methoxy-2-oxindole derivatives showed a pronounced hypotensive effect, while the presence of nitro-groups decreased the hypotensive properties of compounds. The N-unsubstituted and N-benzylated compounds both retain the hypotensive activity, but the most active nitrile **2i** (Figure 2) has both 5-OMe and N-benzyl groups. In comparison with oxindoles from our previous work [19], the advantages of the new compounds are their simpler synthesis and better water solubility, with a comparable hypotensive effect (ΔIOP up to ca. 5 Torr).

Compounds **2h** and **2c**, like timolol and melatonin, showed the maximum effect at 3 h and compound **2i**—4 h after instillation, and all of them represent a more extended hypotensive effect than timolol and melatonin. The more significant hypotensive effects of these compounds (**2c, 2h**, and **2i**) compared to melatonin or timolol 5 h after instillation are statistically significant (Figure 3).

All compounds were well tolerated by the animals. Visual examination revealed no such clinical signs of ocular irritation as conjunctival hyperemia, conjunctival or palpebral edema, lacrimation, or any reactions to discomfort (eye rubbing, avoidance, photophobia).

## 3. Materials and Methods

### 3.1. Chemistry

The 1,4-dioxane was distilled over sodium. The triethylamine and piperidine were dried over NaOH. Reactions were monitored by thin-layer chromatography (TLC) carried out on Merck TLC silica gel plates (60 F254), using UV light for visualization and basic aqueous potassium permanganate or iodine fumes as developing agents. Flash column chromatography purification was performed using silica gel 60 (particle size 0.040–0.060 mm). The melting points were measured in open capillaries and are presented without correction. ^1^H and 13C NMR spectra were recorded at 298 K on a Bruker Avance 400 spectrometer with an operating frequency of 400 and 100 MHz, respectively, and calibrated using residual CHCl_3_ (δH = 7.26 ppm) and CDCl_3_ (δC = 77.16 ppm) or DMSO-d_5_ (δH = 2.50 ppm) and DMSO-d_6_ (δC = 39.52 ppm) as internal references. NMR data are presented as follows: chemical shift (δ ppm), multiplicity (s = singlet, d = doublet, dd = doublet of doublets, t = triplet, q = quartet, m = multiplet, br. = broad), coupling constant (J) in Hertz (Hz), integration. IR spectra were recorded on the Thermo Nicolet IR-200 in KBr or ZnSe. High-resolution mass spectra (HRMS) were measured on a Thermo Scientific LTQ Orbitrap instrument using nanoelectrospray ionization (nano-ESI). HPLC was measured using acetonitrile as the eluent on a Shimadzu Prominence LCMS-2020.

The reactions under microwave irradiation were carried out in a Monowave 300 microwave reactor (Anton Paar, GmbH), as well as in a Bosch HMT72M420 domestic microwave oven with a volume of 17 L. Reactions under thermal activation conditions were carried out on laboratory hot plates, as well as in a ChemiStation chemical reactor (EYELA).

All obtained compounds were sufficiently pure (89–98% based on HPLC and NMR, Appendix A) and could be used in biological activity evaluation and further synthesis without additional purification.

#### 3.1.1. General Procedure for Synthesis of (3-Hydroxy-2-oxo-2,3-dihydro-1H-indol-3-yl)acetonitriles and (3-Hydroxy-2-oxo-2,3-dihydro-1H-indol-3-yl)acetic Acids

Cyanoacetic acid (1 eq.) or malonic acid (2.2 eq.) were added to a solution of isatin derivative **1a**–**i** (1 eq.) in a mixture of dioxane and triethylamine (1.5 eq for malonic acid or 0.25 eq. for cyanoacetic acid). Three types of reaction vessels were used: A chemical test tube with a screw cap (10 mL volume) in method A, a heavy-walled pressure vessel with a PTFE snap cap (30 mL volume) in method B, and a single-necked round-bottom flask (50 mL volume) in method C. The reaction mixture was vigorously stirred until the complete dissolution of the acid.

Method A:

The reaction vessel was placed in a domestic microwave oven under radiation of 360W for 2 min. After that, the reaction vessel was slowly adjusted to ~50 °C, then quickly cooled to room temperature using a cool water bath and again placed in a domestic microwave oven for 2 min at the same power. The cycle of cooling and irradiation was repeated twice more.

Method B:

The mixture in the reaction vessel was stirred with reflux for 3 h.

Method C:

The reaction vessel was placed in a microwave reactor. In the device settings, the temperature was set to 120 °C, the stirring intensity to 600 rpm, and the time to 3 min. Every 3 min, the reaction flask was cooled to room temperature. The cycle of cooling and irradiation was repeated twice more.

The mixture in the reaction vessel was stirred with reflux for 3 h.

Purification (common for all above methods):

The reaction mixture was evaporated to dry in vacuo. The residue was washed with 3 mL of 10% HCl. The product was dissolved in 5 mL EtOAc, dried over Na_2_SO_4_ (anhydrous), and the organic solvent was removed in vacuo. The following compounds had been obtained according to these procedures:

(3-Hydroxy-5-nitro-2-oxo-2,3-dihydro-1H-indol-3-yl)acetonitrile (**2a**) [19].

Briefly, 227 mg (1.18 mmol) of 5-nitro-1H-indole-2,3-dione (**1a**), 0.1 g (1.18 mmol) of cyanoacetic acid, 2 mL of pure dioxane, and 0.1 mL of triethylamine were involved in the reaction under the conditions of methods A and B. After extraction, 211 mg (method A) and 269 mg (method B) of solid product were obtained. Yield 77% (method A) and 98% (method B). NMR spectra are identical.

NMR ^1^H (DMSO-d6): 3.14 (d, J = 16.6, 1H); 3.21 (d, J = 16.7, 1H); 7.12 (br.s., 1H); 8.26 (br.s., 1H); 8.33 (s, 1H).

(3-Hydroxy-7-nitro-2-oxo-2,3-dihydro-1H-indol-3-yl)acetonitrile (**2b**).

Briefly, 227 mg (1.18 mmol) of 7-nitro-1H-indole-2,3-dione (**1b**), 0.1 g (1.18 mmol) of cyanoacetic acid, 2 mL of pure dioxane, and 0.1 mL of triethylamine were involved in the reaction under the conditions of methods A and B. After extraction, 184 mg (method A) and 209 mg (method B) of solid product were obtained. Yield 67% (method A) and 76% (method B). NMR spectra are identical.

NMR ^1^H (DMSO-d6): 3.14 (s, 2H); 7.23–7.31 (m, 1H); 7.85 (d, J = 7.2, 1H); 8.09 (d, J = 8.6, 1H); 11.34 (s, 1H).

(3-Hydroxy-5-methoxy-2-oxo-2,3-dihydro-1H-indol-3-yl)acetonitrile (**2c**) [27].

Briefly, 209 mg (1.18 mmol) of 5-methoxy-1H-indole-2,3-dione (**1c**), 0.1 g (1.18 mmol) of cyanoacetic acid, 2 mL of pure dioxane and 0.1 mL of triethylamine were involved in the reaction under the conditions of methods A and B. After extraction, 154 mg (method A) and 180 mg (method B) of thick oil product were obtained. Yield 60% (method A) and 70% (method B). NMR spectra are identical.

NMR ^1^H (DMSO-d6): 2.94 (d, J = 16.6, 1H); 3.04 (d, J = 16.5, 1H); 3.71 (s, 3H); 6.77 (d, J = 8.4, 1H); 6.85 (dd, J = 8.4, J = 2.7, 1H); 7.08 (d, J = 2.4, 1H); 10.37 (s, 1H).

NMR ^13^C (DMSO-d6): 24.48, 55.54, 72.21, 110.47, 111.14, 114.48, 116.94, 132.37, 134.62, 155.16, 182.85.

(3-Hydroxy-5-methoxy-4-nitro-2-oxo-2,3-dihydro-1H-indol-3-yl)acetonitrile (**2d**).

Briefly, 262 mg (1.18 mmol) of 5-methoxy-4-nitro-1H-indole-2,3-dione (**1d**), 0.1 g (1.18 mmol) of cyanoacetic acid, 2 mL of pure dioxane and 0.1 mL of triethylamine were involved in the reaction under the conditions of methods A and B. After extraction, 133 mg (method A) and 170 mg (method B) of thick oil product were obtained. Yield 43% (method A) and 55% (method B). NMR spectra are identical.

NMR ^1^H (DMSO-d6): 3.09 (s, 2H); 3.84 (s, 3H); 7.03 (br.s, 1H); 7.07 (d, J = 8.6, 1H); 7.28 (d, J = 8.6, 1H); 10.89 (s, 1H).

(3-Hydroxy-5-methoxy-7-nitro-2-oxo-2,3-dihydro-1H-indol-3-yl)acetonitrile (**2e**) [19]

Briefly, 262 mg (1.18 mmol) of 5-methoxy-7-nitro-1H-indole-2,3-dione (**1e**), 0.1 g (1.18 mmol) of cyanoacetic acid, 2 mL of pure dioxane, and 0.1 mL of triethylamine were involved in the reaction under the conditions of methods A and B. After extraction, 248 mg (method A) and 226 mg (method B) of solid product were obtained. Yield 80% (method A) and 73% (method B). NMR spectra are identical. M.p. = 137 °C (m.p.^lit^ = 125–130 °C [19]).

NMR ^1^H (DMSO-d6): 3.10–3.20 (m, 2H); 3.83 (s, 3H); 7.50 (d, J = 2.2, 1H); 7.54 (d, J = 2.4, 1H); 11.22 (s, 1H).

NMR ^13^C (DMSO-d6): 25.68, 56.23, 70.77, 106.18, 116.72, 120.09, 130.94, 132.01, 134.85, 154.48, 176.86.

(3-Hydroxy-5,7-dinitro-2-oxo-2,3-dihydro-1H-indol-3-yl)acetonitrile (**2f**) [19].

Briefly, 280 mg (1.18 mmol) of 5,7-dinitro-1H-indole-2,3-dione (**1f**), 0.1 g (1.18 mmol) of cyanoacetic acid, 2 mL of pure dioxane, and 0.1 mL of triethylamine were involved in the reaction under the conditions of methods A and B. After extraction, 266 mg (method A) and 253 mg (method B) of solid product were obtained. Yield 81% (method A) and 77% (method B). NMR spectra are identical.

NMR ^1^H (DMSO-d6): 3.23–3.34 (m, 2H); 8.62 (br.s, 1H); 8.83 (d, J = 2.2, 1H); 12.03 (s, 1H).

(3-Hydroxy-2-oxo-2,3-dihydro-1H-indol-3-yl)acetonitrile (**2g**) [27].

Briefly, 173 mg (1.18 mmol) of 1H-indole-2,3-dione (**1g**), 0.1 g (1.18 mmol) of cyanoacetic acid, 2 mL of pure dioxane, and 0.1 mL of triethylamine were involved in the reaction under the conditions of methods A and B. After extraction, 146 mg (method A) and 135 mg (method B) of thick oil product were obtained. Yield 66% (method A) and 61% (method B). NMR spectra are identical.

NMR ^1^H (DMSO-d6): 2.94 (d, J = 16.6, 1H); 3.04 (d, J = 16.8, 1H); 6.87 (d, J = 7.8, 1H); 6.99–7.06 (m, 1H); 7.24–7.31 (m, 1H); 7.43–7.50 (m, 1H); 10.56 (s, 1H).

(5-Bromo-3-hydroxy-2-oxo-2,3-dihydro-1H-indol-3-yl)acetonitrile (**2h**) [27].

Briefly, 267 mg (1.18 mmol) of 5-bromo-1H-indole-2,3-dione (**1h**), 0.1 g (1.18 mmol) of cyanoacetic acid, 2 mL of pure dioxane, and 0.1 mL of triethylamine were involved in the reaction under the conditions of methods A and B. After extraction, 227 mg (method A) and 249 mg (method B) of solid product were obtained. Yield 72% (method A) and 79% (method B). NMR spectra are identical.

NMR ^1^H (DMSO-d6): 3.02 (d, J = 16.6, 1H); 3.10 (d, J = 16.6, 1H); 6.83 (d, J = 8.3, 1H); 7.47 (dd, J = 8.3, J = 2.1, 1H); 7.59 (d, J = 1.8, 1H); 10.71 (s, 1H).

NMR ^13^C (DMSO-d6): 25.74, 72.04, 112.15, 113.61, 116.94, 127.09, 132.13, 132.70, 140.98, 176.20.

(1-Benzyl-3-hydroxy-5-methoxy-2-oxo-2,3-dihydro-1H-indol-3-yl)acetonitrile (**2i**)

Briefly, 351 mg (1.18 mmol) of 1-benzyl-5-methoxy-1H-indole-2,3-dione (**1i**), 0.1 g (1.18 mmol) of cyanoacetic acid, 2 mL of pure dioxane, and 0.1 mL of triethylamine were involved in the reaction under the conditions of method A. After extraction, 218 mg of solid product was obtained. Yield 60%.

NMR ^1^H (DMSO-d6): 3.07 (d, J = 16.5, 1H); 3.19 (d, J = 16.6, 1H); 3.71 (s, 3H); 4.86 (s, 2H); 6.78 (d, J = 8.6, 1H); 6.85 (dd, J = 8.6, J = 2.6, 1H); 7.18 (d, J = 2.6, 1H); 7.23–7.36 (m, 5H).

NMR ^13^C (DMSO-d6): 26.17, 42.79, 55.58, 72.20, 110.19, 111.32, 114.23, 116.97, 127.18, 128.56, 135.21, 135.88, 144.58, 155.70, 174.92.

(3-Hydroxy-5-nitro-2-oxo-2,3-dihydro-1H-indol-3-yl)acetic acid (**3a**) [19].

Briefly, 168 mg (0.88 mmol) of 5-nitro-1H-indole-2,3-dione (**1a**), 0.2 g (1.92 mmol) of malonic acid, 3 mL of pure dioxane, and 0.5 mL of triethylamine were involved in the reaction under the conditions of methods A and B. After extraction, 191 mg (method A) and 193 mg (method B) of solid product were obtained. Yield 86% (method A) and 87% (method B).

Briefly, 336 mg (1.75 mmol) of 5-nitro-1H-indole-2,3-dione (**1a**), 0.4 g (3.85 mmol) of malonic acid, 15 mL of pure dioxane, and 1.0 mL of triethylamine were involved in the reaction under the conditions of method C. After extraction, 441 mg of solid product was obtained. Yield 92%.

NMR spectra are identical. M.p. = 212–216 °C (m.p.^lit^ = 215–216 °C [19]).

NMR ^1^H (DMSO-d6): 2.96 (d, J = 16.4, 1H); 3.21 (d, J = 16.5, 1H); 6.40 (br.s, 1H); 6.99 (d, J = 8.6, 1H); 8.18 (dd, J = 8.6, J = 2.4, 1H); 8.26 (d, J = 2.2, 1H); 11.00 (s, 1H).

(3-Hydroxy-7-nitro-2-oxo-2,3-dihydro-1H-indol-3-yl)acetic acid (**3b**) [19].

Briefly, 168 mg (0.88 mmol) of 7-nitro-1H-indole-2,3-dione (**1b**), 0.2 g (1.92 mmol) of malonic acid, 3 mL of pure dioxane, and 0.5 mL of triethylamine were involved in the reaction under the conditions of methods A and B. After extraction, 215 mg (method A) and 206 mg (method B) of solid product were obtained. Yield 97% (method A) and 93% (method B). NMR spectra are identical. M.p. = 241 °C (m.p.^lit^ = 242–243 °C [19]).

NMR ^1^H (DMSO-d6): 3.02 (d, J = 16.4, 1H); 3.12 (d, J = 16.5, 1H); 7.15 (dd, J = 8.6, J = 7.2, 1H); 7.76 (d, J = 7.0, 1H); 8.00 (dd, J = 8.6, J = 1.1, 1H); 11.07 (s, 1H).

(3-Hydroxy-5-methoxy-2-oxo-2,3-dihydro-1H-indol-3-yl)acetic acid (**3c**).

Briefly, 155 mg (0.88 mmol) of 5-methoxy-1H-indole-2,3-dione (**1c**), 0.2 g (1.92 mmol) of malonic acid, 3 mL of pure dioxane, and 0.5 mL of triethylamine were involved in the reaction under the conditions of methods A and B. After extraction, 127 mg (method A) and 104 mg (method B) of solid product were obtained. Yield 61% (method A) and 50% (method B). NMR spectra are identical.

NMR ^1^H (DMSO-d6): 2.86 (d, J = 15.7, 1H); 2.92 (d, J = 15.8, 1H); 3.68 (s, 3H); 6.68 (d, J = 8.5, 1H); 6.75 (dd, J = 8.3, J = 2.5, 1H); 6.96 (d, J = 2.5, 1H); 10.05 (s, 1H).

NMR ^13^C (DMSO-d6): 41.54, 55.49, 73.03, 109.78, 111.20, 113.54, 132.56, 135.95, 154.72, 169.09, 178.06.

HRMS-ESI, m/z: 238.0710 (C_11_H_11_NO_5_, M + H), 260.0529 (C_11_H_11_NO_5_, M + Na), 276.0269 (C_11_H_11_NO_5_, M + K), calculated for C_11_H_11_NO_5_ + H: 238.0710; C_11_H_11_NO_5_ + Na: 260.0529; C_11_H_11_NO_5_ + K: 276.0269.

(3-Hydroxy-5-methoxy-4-nitro-2-oxo-2,3-dihydro-1H-indol-3-yl)acetic acid (**3d**) [19]

Briefly, 194 mg (0.88 mmol) of 5-methoxy-4-nitro-1H-indole-2,3-dione (**1d**), 0.2 g (1.92 mmol) of malonic acid, 3 mL of pure dioxane, and 0.5 mL of triethylamine were involved in the reaction under the conditions of methods A and B. After extraction, 176 mg (method A) and 181 mg (method B) of solid product were obtained. Yield 71% (method A) and 73% (method B). NMR spectra are identical. M.p. = 242 °C (m.p.^lit^ = 240–245 °C [15]).

NMR ^1^H (DMSO-d6): 2.82 (d, J = 16.7, 1H); 2.98 (d, J = 16.6, 1H); 3.79 (s, 3H); 6.49 (br.s, 1H); 6.94 (d, J = 8.6, 1H); 7.15 (d, J = 8.7, 1H); 10.53 (s, 1H).

(3-Hydroxy-5-methoxy-7-nitro-2-oxo-2,3-dihydro-1H-indol-3-yl)acetic acid (**3e**).

Briefly, 194 mg (0.88 mmol) of 5-methoxy-7-nitro-1H-indole-2,3-dione (**1e**), 0.2 g (1.92 mmol) of malonic acid, 3 mL of pure dioxane, and 0.5 mL of triethylamine were involved in the reaction under the conditions of methods A and B. After extraction, 196 mg (method A) and 208 mg (method B) of solid product were obtained. Yield 79% (method A) and 84% (method B). NMR spectra are identical.

NMR ^1^H (DMSO-d6): 2.98 (d, J = 16.7, 1H); 3.17 (d, J = 16.5, 1H); 3.81 (s, 3H); 7.41 (d, J = 2.5, 1H); 7.50 (d, J = 2.3, 1H); 10.94 (s, 1H).

NMR ^13^C (DMSO-d6): 42.29, 56.49, 71.54, 105.34, 113.43, 120.51, 130.66, 134.05, 154.56, 168.95, 170.75.

HRMS-ESI, m/z: 283.0570 (C_11_H_10_N_2_O_7_, M + H), 305.0380 (C_11_H_10_N_2_O_7_, M + Na), calculated for C_11_H_10_N_2_O_7_ + H: 283.0570; C_11_H_10_N_2_O_7_ + Na: 305.0380.

(3-Hydroxy-5,7-dinitro-2-oxo-2,3-dihydro-1H-indol-3-yl)acetic acid (**3f**) [19].

Briefly, 207 mg (0.88 mmol) of 5,7-dinitro-1H-indole-2,3-dione (**1f**), 0.2 g (1.92 mmol) of malonic acid, 3 mL of pure dioxane, and 0.5 mL of triethylamine were involved in the reaction under the conditions of methods A and B. After extraction, 204 mg (method A) and 186 mg (method B) of solid product were obtained. Yield 78% (method A) and 71% (method B). NMR spectra are identical. M.p. = 239 °C (m.p.^lit^ = 235–240 °C [19]).

NMR ^1^H (DMSO-d6): 3.06 (d, J = 17.1, 1H); 3.46 (d, J = 16.9, 1H); 6.70 (br.s, 1H); 8.65 (d, J = 1.9, 1H); 8.78 (d, J = 2.3, 1H); 11.79 (s, 1H).

(3-Hydroxy-2-oxo-2,3-dihydro-1H-indol-3-yl)acetic acid (**3g**) [26].

Briefly, 129 mg (0.88 mmol) of 1H-indole-2,3-dione (**1g**), 0.2 g (1.92 mmol) of malonic acid, 3 mL of pure dioxane, and 0.5 mL of triethylamine were involved in the reaction under the conditions of methods A and B. After extraction, 138 mg (method A) and 127 mg (method B) of solid product were obtained. Yield 76% (method A) and 70% (method B).

Briefly, 264 mg (1.80 mmol) of 1H-indole-2,3-dione (**1g**), 411 mg (3.95 mmol) of malonic acid, 15 mL of pure dioxane, and 1.0 mL of triethylamine were involved in the reaction under the conditions of method C. After extraction, 286 mg of solid product was obtained. Yield 77%.

NMR spectra are identical.

NMR ^1^H (DMSO-d6): 2.51 (d, J = 15.9, 1H); 2.61 (d, J = 15.4, 1H); 6.75 (d, J = 7.7, 1H); 6.86–6.92 (m, 1H); 7.11–7.17 (m, 1H); 7.27 (d, J = 7.2, 1H); 10.17 (br.s, 1H).

NMR ^13^C (DMSO-d6): 45.28, 73.23, 109.40, 121.27, 123.76, 128.77, 132.52, 142.21, 172.35, 178.46.

(5-Bromo-3-hydroxy-2-oxo-2,3-dihydro-1H-indol-3-yl)acetic acid (**3h**).

Briefly, 198 mg (0.88 mmol) of 5-bromo-1H-indole-2,3-dione (**1h**), 0.2 g (1.92 mmol) of malonic acid, 3 mL of pure dioxane, and 0.5 mL of triethylamine were involved in the reaction under the conditions of methods A and B. After extraction, 194 mg (method A) and 202 mg (method B) of solid product were obtained. Yield 77% (method A) and 80% (method B). NMR spectra are identical.

NMR ^1^H (DMSO-d6): 2.89 (d, J = 16.1, 1H); 3.02 (d, J = 16.1, 1H); 6.74 (d, J = 8.2, 1H); 7.36 (dd, J = 8.2, J = 1.5, 1H); 7.51 (s, 1H); 10.37 (s, 1H).

NMR ^13^C (DMSO-d6): 41.38, 72.61, 111.46, 113.01, 126.94, 131.75, 133.87, 142.14, 170.31, 177.67.

HRMS-ESI, m/z: 285.9709 (C_10_H_9_BrNO_4_, M + H), 307.9529 (C_10_H_9_BrNO_4_, M + Na), calculated for C_10_H_9_BrNO_4_ + H: 285.9709; C_10_H_9_BrNO_4_ + Na: 307.9529.

(1-Benzyl-3-hydroxy-5-methoxy-2-oxo-2,3-dihydro-1H-indol-3-yl)acetic acid (**3i**).

Briefly, 237 mg (0.88 mmol) of 1-benzyl-5-methoxy-1H-indole-2,3-dione (**1i**), 0.2 g (1.92 mmol) of malonic acid, 3 mL of pure dioxane, and 0.5 mL of triethylamine were involved in the reaction under the conditions of method A. After extraction. 161 mg of solid product was obtained. Yield 56%.

NMR ^1^H (DMSO-d6): 3.01 (d, J = 15.9, 1H); 3.07 (d, J = 15.9, 1H); 3.67 (s, 3H); 4.74 (d, J = 16.0, 1H); 4.89 (d, J = 16.0, 1H); 6.59 (d, J = 8.4, 1H); 6.73 (dd, J = 8.4, J = 2.6, 1H); 7.05 (d, J = 2.6, 1H); 7.20–7.32 (m, 3H); 7.38 (d, J = 7.1, 1H).

NMR ^13^C (DMSO-d6): 41.63, 42.81, 55.54, 66.40, 72.74, 109.40, 111.21, 113.25, 127.18, 128.45, 131.97, 136.40, 136.57, 155.35, 170.39, 176.40.

#### 3.1.2. Procedure for Synthesis of (3-Hydroxy-2-oxo-2,3-dihydro-1H-indol-3-yl)malonic Acid (X) [26]

A chemical test tube with a screw cap containing a reaction mixture was placed in a Bosch HMT72M420 domestic microwave oven (17 L volume) under radiation of 360W for 2 min. After that, the reaction vessel was cooled to room temperature. Briefly, 129 mg (0.88 mmol) of 1H-indole-2,3-dione (**1g**), 0.3 g (2.88 mmol) of malonic acid, 3 mL of pure dioxane, and 0.5 mL of triethylamine were involved in the reaction. A chemical test tube with a screw cap containing a reaction mixture was placed in a Bosch HMT72M420 domestic microwave oven (17 L volume) under radiation of 360W for 2 min. After that, the reaction vessel was cooled to room temperature. The reaction mixture was poured hot into a single-necked round-bottom flask, and the organic solvent was removed in vacuo. The resulting compound was washed with small portions of 10% HCl. The product was dissolved in 5 mL EtOAc, dried over Na_2_SO_4_ (anhydrous), and the organic solvent was removed in vacuo. One hundred and sixty-six milligrams of solid product was obtained. Yield 75%.

NMR ^1^H (DMSO-d6): 3.55 (s, 1H); 6.02 (br.s, 1H); 6.76 (d, J = 7.7, 1H); 6.88–6.94 (m, 1H); 7.13–7.21 (m, 1H); 7.29 (d, J = 7.3, 1H); 10.21 (s, 1H).

NMR ^13^C (DMSO-d6): 41.97, 72.64, 109.50, 121.36, 123.96, 127.04, 129.20, 142.81, 168.51, 168.54, 170.33.

### 3.2. In Vivo Testing

The ability of oxindoles to reduce IOP was assessed on normotensive male Chinchilla rabbits weighing about 2 kg. All investigated substances were used topically as 0.1% solutions (*w*/*v*); in the case of melatonin, this corresponded to a concentration of 4.3 μM. All substances were dissolved in a 0.05 M phosphate buffer solution, pH 7.4, containing 5% dimethyl sulfoxide (DMSO) *v*/*v*. All compounds were initially dissolved in DMSO and then diluted to the required concentration with phosphate buffer solution. The solutions were instilled into both eyes at a fixed volume of 50 μL. IOP was measured before instillation and then followed for 6 h at a 1 h interval using an automatic tonometer for veterinary Tonovet (Icare, Finland). Evaluation of the hypotensive effect of each compound was carried out in a group of 5 animals (10 eyes). Results were corrected for normal daily IOP fluctuations by simultaneous IOP measurement in 3 intact animals (6 eyes) instilled with a phosphate buffer solution containing 5% DMSO. Animals were taken for the next experiment after at least 7 days. Since the rate of IOP decrease varied in different animals during the study, we took into account not only the average decrease in the group at a particular time point but also the maximal IOP decrease in each eye.

## 4. Conclusions

The microwave-assisted (MW) decarboxylative condensation of isatins with malonic and cyanoacetic acids is reported to give the new 3-hydroxy-2-oxindole derivatives with high yields (up to 98%). The obtained compounds were found to reduce the intraocular pressure (IOP) in normotensive rabbits nearly as effectively as melatonin and timolol (reference drugs) or even more effectively: the lead compound was found to reduce the IOP by 5.6 Torr (ΔIOP for the timolol 3.50 Torr and for melatonin 2.7 Torr). A time-dependent study revealed the prolonged effects (more than 5 h) of the synthesized compounds. This hypotensive effect without any signs of local ocular toxicity in combination with high water solubility represents a great potential for low-cost oxindole derivatives as potent antiglaucomatousic agents. The results of this pilot study suggest the prospects for further preclinical studies of the efficacy and safety of the identified leading compounds as potential antiglaucomatous drugs.

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
