# Peer review of "Microwave-Assisted Synthesis of 3-Hydroxy-2-oxindoles and Pilot Evaluation of Their Antiglaucomic Activity"

_ijms, 2023, doi:10.3390/ijms24065101_

Round 1

Reviewer 1 Report (Previous Reviewer 1)

In this manuscript, an efficient method for obtaining novel 2-oxindole derivatives via MW decarboxylative condensation of substituted isatins with malonic and cyanoacetic acids is reported. Two mechanisms have been proposed and the influence of novel compounds applied in instillations on IOP was studied in vivo on normotensive rabbits. The manuscript is well written, but the authors must address some concerns before it is considered for publication.

1. Line 21, should the term "antiglaucomatousic" be "antiglaucomatous"? I did not find anything about "antiglaucomatousic" online.

2. Line 171, you mentioned “all obtained compounds were sufficiently pure (≥95%)”, which method(s) did you use for the purity check? If you used HPLC, please add the chromatograms in the manuscript.

3. Line 185, you mentioned “After then the reaction vessel was cooled to room temperature”, did you cool down to room temp quickly or slowly?

Author Response

First of all, we would like to thank the reviewer for careful analysis of our article and the questions asked. Below are answers to questions and a description of the changes.

In this manuscript, an efficient method for obtaining novel 2-oxindole derivatives via MW decarboxylative condensation of substituted isatins with malonic and cyanoacetic acids is reported. Two mechanisms have been proposed and the influence of novel compounds applied in instillations on IOP was studied in vivo on normotensive rabbits. The manuscript is well written, but the authors must address some concerns before it is considered for publication.

  1. Line 21, should the term "antiglaucomatousic" be "antiglaucomatous"? I did not find anything about "antiglaucomatousic" online.

Answer: Thank you – this is erratum. We replaced it with antiglaucomatous

  1. Line 171, you mentioned “all obtained compounds were sufficiently pure (≥95%)”, which method(s) did you use for the purity check? If you used HPLC, please add the chromatograms in the manuscript.

Answer:   

The purity of obtained compounds was evaluated by 1H-NMR and by HPLC. Additional HPLC data were added in supplementary information.

  1. Line 185, you mentioned “After then the reaction vessel was cooled to room temperature”, did you cool down to room temp quickly or slowly?

Answer:        

Reaction mixture was slowly adjusted to ~50C, then quickly cooled to room temperature using cool water bath. Line 197 was corrected respectively.

Reviewer 2 Report (Previous Reviewer 2)

Major concerns:

1.Fig2 and results:The decreasing intraocular pressure comparison among 3-hydroxy-2-oxindoles , Timopol and melatonin should have a stastistic comparison. Line 129 but most active nitrile 2i has both 5-OMe  and N-benzyl groups (figure 2).

2. Fig.3 Add the number of rabbits tested in each group.

Overall, PI have found the effective ways to produce and decrease intraocular pressure in normal rabbits. However, this manuscript lacks of convincing data to demonstrate this new drug is better than the clinical decreasing intraocular pressure (iop) drugs used in treating glaucoma. Timoptol is not the stronggist drug to decreasing iop, a prostaglandin drug should be compared in rabbits with high iop.

Author Response

First of all, we would like to thank the reviewer for careful analysis of our article and the questions asked. Below are answers to questions and a description of the changes.

Reviewer 2

Major concerns:

1.Fig2 and results:The decreasing intraocular pressure comparison among 3-hydroxy-2-oxindoles , Timopol and melatonin should have a stastistic comparison. Line 129 but most active nitrile 2i has both 5-OMe  and N-benzyl groups (figure 2).

Answer:

We include the data about the statistical comparisons of ∆IOP to figure 2 and caption to this figure.

  1. Fig.3 Add the number of rabbits tested in each group.

Answer: All compounds were tested on 5 animals (10 eyes) and this data also was included to caption to fig.3.

Overall, PI have found the effective ways to produce and decrease intraocular pressure in normal rabbits. However, this manuscript lacks of convincing data to demonstrate this new drug is better than the clinical decreasing intraocular pressure (iop) drugs used in treating glaucoma. Timoptol is not the stronggist drug to decreasing iop, a prostaglandin drug should be compared in rabbits with high iop.

Answer:

We consider that testing on normotensive animals is enough to estimate the very ability of a substance to lower the IOP – so that it is a common method for the drug screening. Current investigation is in progress and comparison of biological effect of 3-hydroxy-2-oxindoles in different animal models is one of the following steps.

Our standard animal model for in vivo assays is normotensive rabbit. It is well known that prostaglandin analogues do not affect rabbit’s IOP due to the absence of trabecular meshwork. We tested Xalatan in rabbits and it didn't change IOP while beta-blockers acted the same as they did in humans.

Reviewer 3 Report (New Reviewer)

The authors have developed a series of 3-hydroxy-2-oxindoles using microwave-assisted synthesis. The synthesized compounds were then conveyed in an animal study to evaluate their anti-glaucoma effects. Three compounds from their synthesis have suggested better effects on decreasing the IOP than melatonin and timolol. The result is interesting. However, I would like the authors to address the following issues before the manuscript can be published. 

(1) On line 84, the author mentioned that the optimization of reaction conditions was performed, but there is no actual data is revealed. The optimization process of microwave-assisted synthesis should be shown. How much do other solvents like THF and EtOH affect? And what about other equivalents of masonic acid and cyanoacetic acid? 

(2) Reference 19 "Oxindole-based intraocular pressure reducing agents" was mentioned on line 118. Please discuss the comparison between the hydroxy-2-oxindole and 2-oxindole derivatives mentioned in the cited paper in the same way. 

(3) The error bars in figure 2 miss the low bar. 

(4) Need more discussion about figure 3. Is there any possible reason why using timolol for comparison instead of melatonin? And why does the initial IOP vary so much (about 20%) when administrated drugs? It may be better to use percentage decreases to show the change. 

(5) As described in the method section 3.2, "the rate of IOP decrease varied in different animals", please show detailed data on how each compound affects IOP. 

(6) The IOP decreased by 5.6 Torr on a 16 Torr basis using the lead compound. It is a roughly 30% decrease. What is an appropriate range for IOP? Will the significant decline cause any issues? It would be great if this would be discussed as well. 

Author Response

First of all, we would like to thank the reviewer for careful analysis of our article and the questions asked. Below are answers to questions and a description of the changes.

Reviewer 3

The authors have developed a series of 3-hydroxy-2-oxindoles using microwave-assisted synthesis. The synthesized compounds were then conveyed in an animal study to evaluate their anti-glaucoma effects. Three compounds from their synthesis have suggested better effects on decreasing the IOP than melatonin and timolol. The result is interesting. However, I would like the authors to address the following issues before the manuscript can be published. 

(1) On line 84, the author mentioned that the optimization of reaction conditions was performed, but there is no actual data is revealed. The optimization process of microwave-assisted synthesis should be shown. How much do other solvents like THF and EtOH affect? And what about other equivalents of masonic acid and cyanoacetic acid? 

Answer:

Table of optimization of microwave-assisted synthesis was added (see Table 2, line 84)

(2) Reference 19 "Oxindole-based intraocular pressure reducing agents" was mentioned on line 118. Please discuss the comparison between the hydroxy-2-oxindole and 2-oxindole derivatives mentioned in the cited paper in the same way. 

Answer:

The comparison new effective compounds with our previously synthesized compounds was added (lines 135-137).

(3) The error bars in figure 2 miss the low bar. 

Answer: We corrected the inaccuracies and made changes to figure 2, including the statistical significance analysis data

Round 2

Reviewer 2 Report (Previous Reviewer 2)

No further comment

This manuscript is a resubmission of an earlier submission. The following is a list of the peer review reports and author responses from that submission.

Round 1

Reviewer 1 Report

In this manuscript, an efficient method for obtaining novel 2-oxindole derivatives via MW decarboxylative condensation of substituted isatins with malonic and cyanoacetic acids is reported. Two mechanisms have been proposed and the influence of novel compounds applied in instillations on IOP was studied in vivo on normotensive rabbits. The manuscript is well written, but the authors must address some concerns before it is considered for publication.

1. Line 17, a typo – decarboxylation instead of decaboxylation

2. Line 29, could you add some references to prove “The only proven way of therapy is a life-long daily use of hypotensive drugs to slow down the progression of optic nerve damage”?

3. Could you please add the purity of your final product?

Author Response

First of all, we would like to thank the reviewer for careful analysis of our article.

Below are answers to reviewer’s questions and a description of the changes. We also changed the appearance of the drawings (which does not change their meaning) and supplemented their description.

Reviewer 2 Report

In this manuscript, Efremos et al., used a novel  2-oxindole derivatives via microwave-assisted (MW) decaboxylative condensation of substituted isatins with malonic and cyanoacetic acids to demonstrate their decreasing intraocular pressure in normotensive rabbits. It is interesting but several issue should be addressed.

1. The writing structure and figure arrangement are not follow the instruction provided by the Journal, and there is no conclusion.

2. Several anti-glaucomatous eyedrops are already available in the clinic.  Authors should provide the rationale to use the 2-oxindole derivatives as a novel treatment of glaucoma. Any evidenes to support that  it would be better than other anti-glaucomatous agents?

3. The most powerful iop reduction drugs is prostaglandin analogues, why authors just pick beta-blocker to compare?

4. The mechanisms of 3-hydroxy-2-oxindoles to reducing iop should be introduced in the introduction section.

5. Authors should provide the comparison between 3-hydroxy-2-oxindoles and current anti-glaucomatous agents to test the efficacy and safety issue in glaucomatous animal models, not only in normotensive rabbits.  

Author Response

(The authors gave the same response as above.)
